# Transition metal migration and $O_2$ formation underpin voltage hysteresis in oxygen-redox disordered rocksalt cathodes

Kit McColl [1,2], Robert A. House [2,3], Gregory J. Rees [2,3], Alexander G. Squires[1], Samuel W. Coles [1,2], Peter G. Bruce [2,3,4], Benjamin J. Morgan[1,2] & M. Saiful Islam [1,2,3] ✉

Lithium-rich disordered rocksalt cathodes display high capacities arising from redox chemistry on both transition-metal ions (TM-redox) and oxygen ions (O-redox), making them promising candidates for next-generation lithium-ion batteries. However, the atomic-scale mechanisms governing O-redox behaviour in disordered structures are not fully understood. Here we show that, at high states of charge in the disordered rocksalt $Li_2MnO_2F$, transition metal migration is necessary for the formation of molecular $O_2$ trapped in the bulk. Density functional theory calculations reveal that $O_2$ is thermodynamically favoured over other oxidised O species, which is confirmed by resonant inelastic X-ray scattering data showing only $O_2$ forms. When O-redox involves irreversible Mn migration, this mechanism results in a path-dependent voltage hysteresis between charge and discharge, commensurate with the hysteresis observed electrochemically. The implications are that irreversible transition metal migration should be suppressed to reduce the voltage hysteresis that afflicts O-redox disordered rocksalt cathodes.

The global uptake of electric vehicles is driving demand for lithium-ion batteries with greater energy densities[1], hence the need for new cathodes with higher capacities[2]. Lithium-rich cathode materials with lithium:transition metal (TM) ratios >1, including layered $Li_{1+x}(Ni,Mn,Co)_{1-x}O_2$ and disordered rocksalts such as $Li_2MnO_2F$, offer increased capacities over conventional cathodes such as $LiCoO_2$ and $LiFePO_4$[3–8]. These high capacities are possible because Li-rich cathodes can exhibit reversible redox of bulk oxide ions, termed 'oxygen-redox', as well as transition metal ion redox[9–11]. O-redox allows Li-rich cathode materials to achieve theoretical capacities exceeding 300 mA h g$^{-1}$ ref. 9, which opens up a new frontier in battery chemistry.

However, one critical issue is that O-redox is almost always associated with a large voltage hysteresis in the first-cycle electrochemical load curve[12–17]. The degree to which fundamental atomic-scale mechanisms of O-redox in Li-rich cathodes contribute to voltage hysteresis is not fully understood and remains a topic of considerable debate[18–32]; in particular the nature of the oxidised O species formed on charge and the role of TM rearrangements are unclear[33]. To harness Li-rich cathodes for technological use, the interrelations between O-redox, TM migration and voltage hysteresis must be fully understood so that strategies can be found to mitigate the loss of energy density.

The disordered rocksalt cathode $Li_2MnO_2F$ exhibits a large capacity, comparable to that of Li-rich ordered layered oxides[34,35]. Previous studies using resonant inelastic X-ray scattering (RIXS) and density functional theory (DFT) have identified molecular $O_2$ formed and trapped within the bulk structure when charged to 4.8 V (approximately $Li_{0.75}MnO_2F$)[35]. $Li_2MnO_2F$ and some other disordered rocksalts[6,34,36–38] display a smaller first cycle voltage hysteresis than ordered layered cathode counterparts[10,39], which raises important questions: what is the atomic-scale O-redox mechanism within these disordered materials, and how does their local structure facilitate a more reversible O-redox process than in Li-rich layered cathode

[1]Department of Chemistry, University of Bath, Bath, UK. [2]The Faraday Institution, Harwell Science and Innovation Campus, Didcot, UK. [3]Department of Materials, University of Oxford, Oxford, UK. [4]Department of Chemistry, University of Oxford, Oxford, UK. ✉e-mail: saiful.islam@materials.ox.ac.uk

materials? In the highly ordered Li-rich layered cathodes, it is established that TM migration is necessary for $O_2$ formation[14,16,21,22,40,41]. In disordered rocksalt materials such as $Li_2MnO_2F$, however, it is still unclear what role, if any, TM migration plays in $O_2$ formation.

To address these questions about transition metal migration, $O_2$ formation and voltage hysteresis in $Li_2MnO_2F$, we have conducted a multi-technique study of the charge-storage mechanism using DFT and ab initio molecular dynamics (AIMD) simulations, high-resolution RIXS mapping and galvanostatic intermittent titration technique (GITT) electrochemical measurements. Our DFT results reveal that in the highly charged (delithiated) state, molecular $O_2$ species are thermodynamically favoured over superoxide and peroxide species. This result is confirmed by new high-resolution RIXS mapping data, which shows vibrational features from molecular $O_2$ only, with no evidence for superoxide and peroxide species. Using AIMD, we resolve an O–O dimerisation mechanism that involves TM migration and features peroxide and superoxide species as short-lived (picosecond timescale) reaction intermediates, before ultimately forming the thermodynamic endproduct, molecular $O_2$. We implicate irreversible Mn migration as a contributor to first cycle voltage hysteresis in $Li_2MnO_2F$ and discuss how fully-reversible Mn migration could provide a route to stable O-redox cycling through $O_2$ formation without voltage loss.

## Results

### Oxygen environments and short-range order in $Li_2MnO_2F$

In disordered rocksalt-structured $Li_2MnO_2F$, octahedrally coordinated cations (Li, Mn) and anions (O, F) occupy two interpenetrating face-centred cubic (fcc) sublattices (Fig. 1a). Disordered rocksalts do not display long-range order, but do exhibit short-range cation order; i.e., preferential local structural motifs[37,42–46]. Understanding how short-range order affects the local structure around oxygen anions in $Li_2MnO_2F$ is important because O-redox activity in Li-rich cathodes has previously been attributed to the preferential oxidation of specific lattice $O^{2-}$ ions with Li-rich coordination environments[47–49]. O

ions with a higher number of Li neighbours have a lower Madelung site potential, which indicates a lower energy required to localise an electron hole ($O^{2-} \rightarrow O^- + e^-$)[49], while oxygen ions with linear Li–O–Li bonding configurations have O 2p states at the top of the valence band that are susceptible to oxidation on charge[47,48]. Furthermore, the possible presence of extremely lithium-rich oxygen-coordination (i.e. $O–Li_6$) is of particular interest, because removal of these Li during charging may leave these O undercoordinated with no directly bonded Mn neighbours, and potentially allowing O–O dimerisation without requiring Mn–O bond breaking or Mn migration.

We first quantify the frequency of different oxygen coordination environments ($O–Li_6$, $O–Li_5Mn$, etc) in pristine, as-prepared $Li_2MnO_2F$. We used DFT calculations to parameterise a cluster-expansion Hamiltonian to describe the short-range interactions in $Li_2MnO_2F$ and ran lattice Monte Carlo simulations at $T = 2000$ K to approximate the ball-milling synthesis conditions of the pristine, as-prepared material[34] (for computational details, see Methods section). Figure 1b shows the predicted frequencies of different O-ion coordination octahedra, $O–Li_xMn_{6-x}$, obtained by sampling structures from these simulations. We also show data for a model at $T = \infty$ K, which represents a hypothetical fully random arrangement of the rocksalt lattice, i.e., with no short-range order (Supplementary Note S2.1).

The fully random ($T = \infty$ K) model of $Li_2MnO_2F$ features a binomial distribution of $O–Li_xMn_{6-x}$ (Fig. 1b), which is skewed towards O-octahedra with a high number of Li neighbours due to the 2:1 ratio of Li:Mn in the material. In this fully random model, 8.5% of the O-environments are $O–Li_6$. In the system approximating the ball-milling synthesis conditions ($T = 2000$ K), the distribution of O-environments deviates from a fully random binomial distribution (Fig. 1b), which indicates short-range order in pristine $Li_2MnO_2F$. $O–Li_6$ environments are predicted to have a very low abundance (<0.05%), and F-ions preferentially occupy anion sites with high numbers of Li neighbours (Supplementary Fig. 1)[43,45]. Because oxygen ions in $O–Li_6$ coordination appear with a very low frequency in the pristine material, the molecular $O_2$ that is observed in experiments upon cycling[34] cannot originate uniquely from starting $O–Li_6$ oxygen ion sites. Instead, the $O_2$ molecules must arise from $O–Li_nMn_{6-n}$ (where $n \leq 5$) sites in the pristine material, or $O–Li_x\square_{6-x}$ configurations (where $\square$ is a vacancy) that could form during charge due to O, Li or Mn displacement.

### Stable structures and nature of oxidised oxygen on lithium extraction

Having characterised the anion short-range order in pristine $Li_2MnO_2F$, we now investigate the charge mechanism by examining structures of highly delithiated $Li_{0.67}MnO_2F$. This stoichiometry corresponds to structures charged past the limit of $Mn^{3+}$ to $Mn^{4+}$ redox and provide new insights into the thermodynamics of different oxidised O species and the possibility of Mn migration.

First, we perform a random structural search at a stoichiometry of $Li_{0.67}MnO_2F$ by generating 150 rocksalt configurations with random distributions of anions and cations, plus cation vacancies, over their respective sublattices. This random structure search allows an unbiased sampling of all hypothetical rocksalt-structured configurations of $Li_{0.67}MnO_2F$, without imposing any conditions on the pristine $Li_2MnO_2F$ structure or on the kinetic pathway that might be required to reach these delithiated structures in experimental samples. The random structure search therefore generated structures that could be obtained under unrestricted TM and anion rearrangement during charge; we label this the 'Mn-rearrangement' model. Second, we model delithiation of the pristine material, under the condition that no TM ion migration is permitted, denoted as the 'constrained-Mn' model. Here, we used our cluster expansion model at $T = 2000$ K to generate 150 $Li_2MnO_2F$ structures, representative of the pristine material. These structures were delithiated either (i) by removing random lithium ions or (ii) removing Li based on a ranking of site energies from

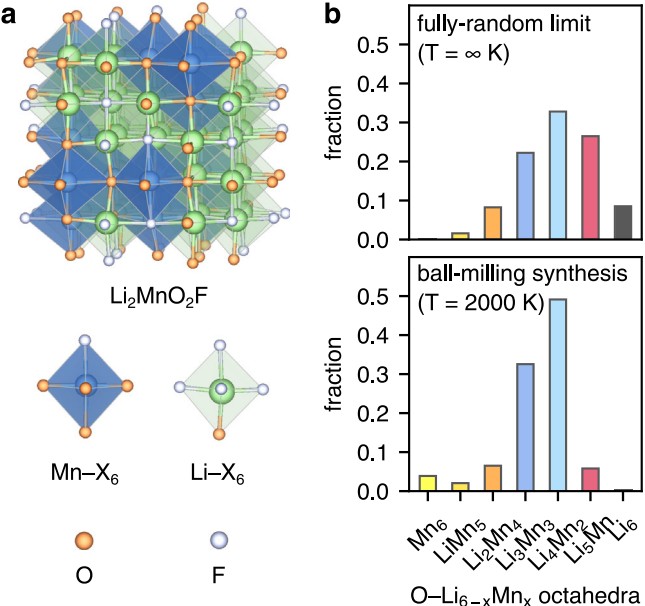

**Fig. 1 | Structure and short-range order of $Li_2MnO_2F$. a** Representative relaxed structure of disordered $Li_2MnO_2F$ in a (2 × 2 × 2) expansion of the conventional rocksalt unit cell. **b** Frequency of $O–Li_xMn_{6-x}$ octahedra in $Li_2MnO_2F$ at $T = \infty$ K (upper panel) derived from a binomial distribution with $n = 6$, $p = 2/3$, representing the fully-random limit and at $T = 2000$ K (lower panel) obtained from cluster-expansion based Monte Carlo simulations, representing the pristine 'as-prepared' material.

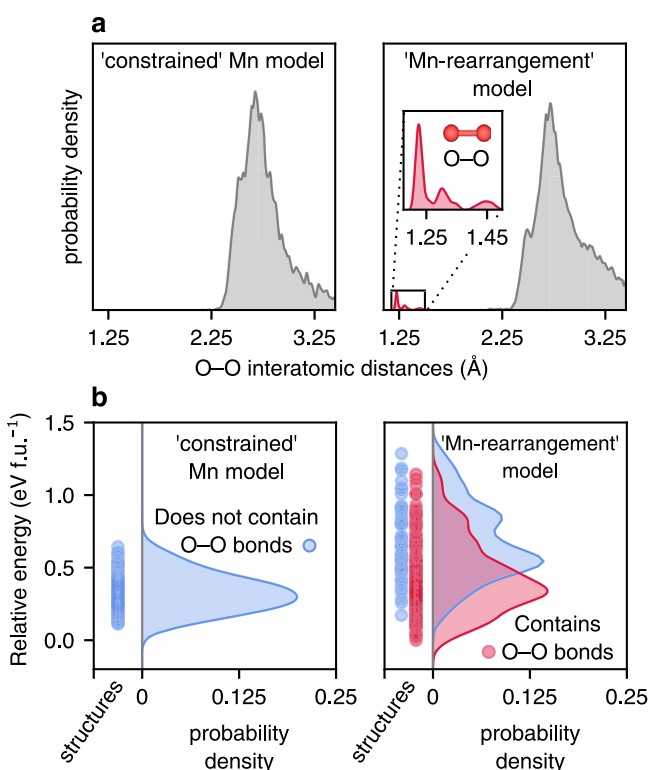

**Fig. 2 | Search for O–O bonds and thermodynamics of O-dimer formation in delithiated $Li_{0.67}MnO_2F$. a** Search for O–O dimers by an analysis of O–O interatomic distances in delithiated structures of the pristine material obtained from the cluster-expansion ('constrained' Mn), and in 'Mn rearrangement' models.
**b** Energetics of the structures from **a** with the structures from the 'Mn-rearrangement' model separated into those containing O–O bonds (<1.7 Å) and those not containing O–O bonds. The right section of each panel shows the kernel density estimations of the probability of energies in the left panel, where each dot is the energy of one structure.

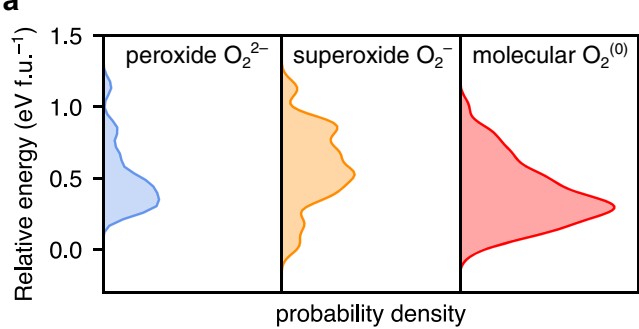

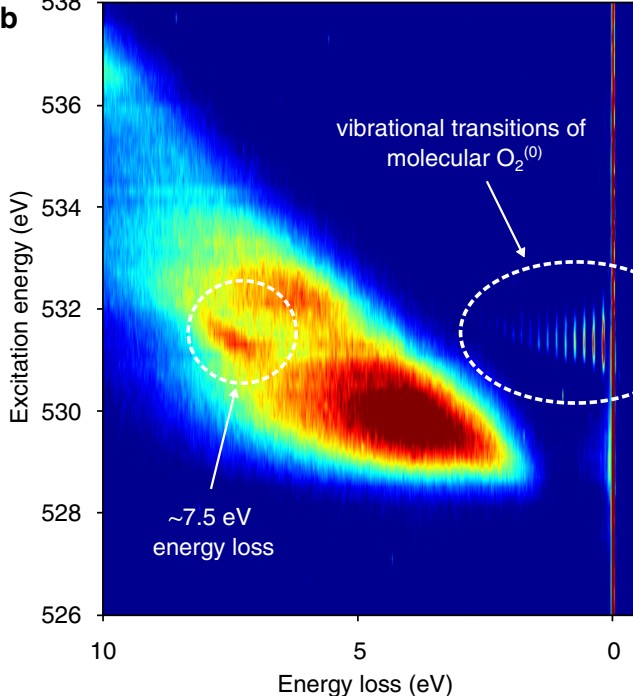

**Fig. 3 | Computational and experimental evidence for molecular $O_2$ formation in charged $Li_{2−x}MnO_2F$. a** Comparison of the stability of structures from the Mn rearrangement model, containing the different types of O–O dimers, classified according to bond length: (peroxide $O_2^{2-}$ (1.35 Å $\leq d_{O-O}$ < 1.70 Å), superoxide $O_2^-$ (1.24 Å $\leq d_{O-O}$ < 1.35 Å) and molecular $O_2$ ($d_{O-O}$ < 1.24 Å). The panels show the kernel density estimations of the probability of energies for the classified structures. The energies were calculated relative to the most stable configuration from the entire search. Thus, zero energy corresponds to the most stable obtained from the Mn rearrangement model. Calculations used the SCAN functional. **b** High-resolution resonant inelastic X-ray scattering (RIXS) map of $Li_{2−x}MnO_2F$ charged to 5.0 V, showing the vibrational features corresponding to molecular $O_2$ only, and an energy loss feature at ~7.5 eV.

electrostatics, to reach a composition of $Li_{0.67}MnO_2F$ and then relaxed with DFT. The relaxations quench the structures to a local potential energy minimum and do not allow for significant atomic rearrangements such as Mn migration.

To assess whether O–O dimerisation occurs in any of the structures in either of the two models, we consider the distributions of O–O distances in the relaxed structures (Fig. 2a); distances shorter than 1.7 Å are indicative of covalent O–O dimerisation. In the constrained-Mn model structures, we find no O-dimers of any kind. Charge compensation beyond the $Mn^{4+}$ limit in these models is predominantly from lattice $O^{n-}$ ions. In contrast, a large proportion of structures from the Mn-rearrangement model show some O–O interatomic distances <1.7 Å, indicating short covalent O–O bonds (Fig. 2a).

Insight into the thermodynamics of O–O dimerisation in $Li_{0.67}MnO_2F$ is provided by considering the calculated energies of all the relaxed structures for the constrained Mn and Mn-rearrangement models (Fig. 2b). For the Mn-rearrangement model, structures that contain covalent O–O bonds are on average ~0.4 eV per formula unit more stable than those that do not contain covalent O–O bonds. Furthermore, the lowest energy structures across both models contain covalent O–O bonds. The results therefore suggest a thermodynamic driving force may exist for pristine $Li_2MnO_2F$ to undergo a framework transformation upon delithiation to allow O–O dimerisation.

The lack of dimerisation in the constrained Mn model is because O–O dimerisation is an activated process that requires either Mn–O bond breaking, or Mn/O displacement. All O atoms in the constrained Mn model start with at least one Mn neighbour, and the structures

relax to the nearest local energy minimum, rather than overcoming the barrier needed for O–O dimerisation. In contrast, in the Mn rearrangement model, some O atoms begin with no Mn neighbours and can dimerise without an activation barrier, so covalent O–O bonds ($d_{O-O}$ < 1.7 Å) form during the geometry relaxations.

There is an ongoing debate over the bond length and oxidation state of O–O dimers in charged O-redox cathodes; peroxide $O_2^{2-}$ ($d_{O-O}$ ~1.44 Å) or $(O \cdots O)^{n-}$ species with long interatomic separations of ~2.4 Å are sometimes invoked to explain capacity from oxidised $O^{[11,21,50]}$. In relaxed structures from the random structure search, the probability density for covalent O–O bonds has maxima at ~1.22 Å, ~1.30 Å and ~1.45 Å, (Fig. 2a, inset) indicating that molecular $O_2^{(0)}$, superoxide $O_2^-$ and peroxide $O_2^{2-}$ species could all form in $Li_{0.67}MnO_2F$. In Fig. 3a, we

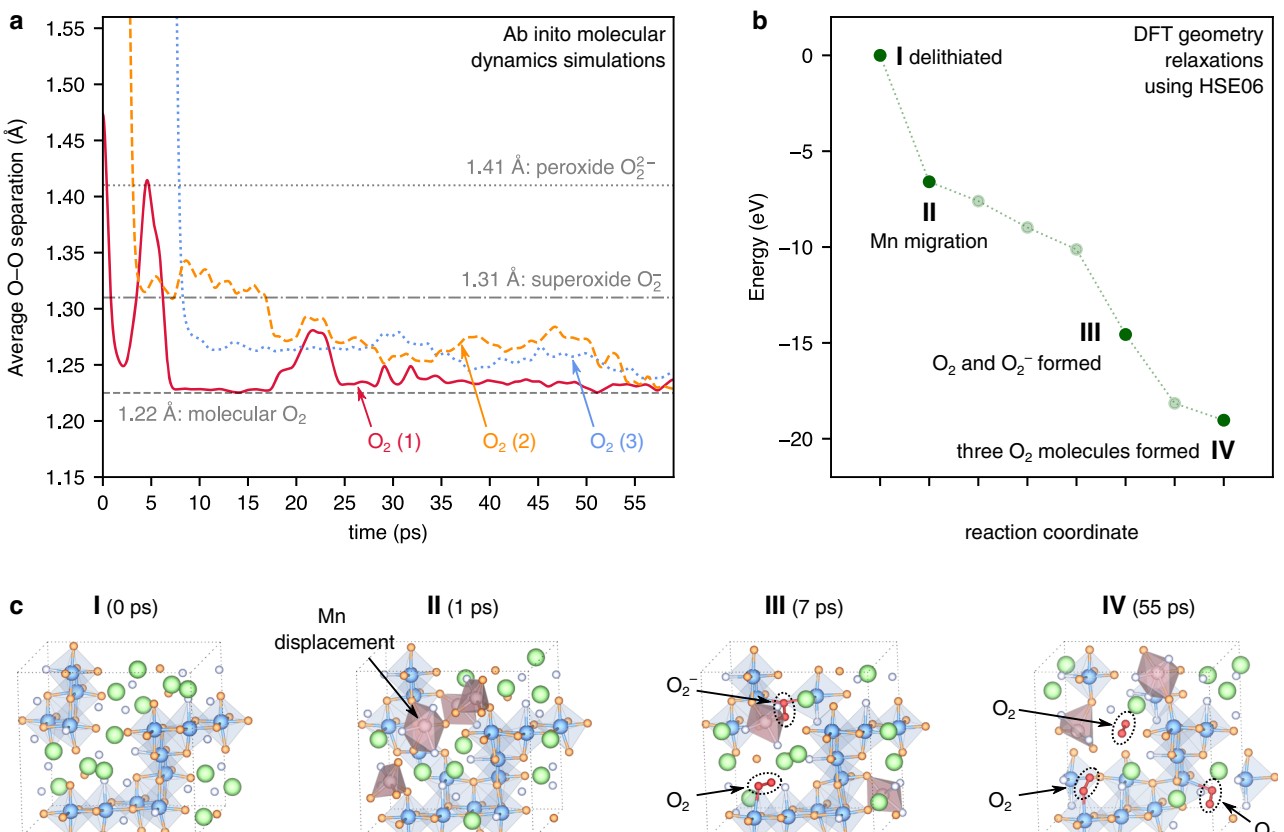

**Fig. 4 | Reaction mechanism to form O−O dimers in Li$_{0.67}$MnO$_2$F from ab inito molecular dynamics (AIMD) and geometry relaxations. a** O···O interatomic separation of O species forming O−O dimers from GGA+$U$ AIMD simulations at 500 K. **b** Total energy of selected structures along the AIMD trajectory, fully relaxed with DFT using the HSE06 functional. **c** Geometry of the relaxed structures from **b**, indicating Mn migration events and O−O dimerisation. Mn ions displaced from their octahedrally coordinated rocksalt sites are indicated by pink polyhedra.

compare the thermodynamic stability of structures containing the different types of O−O dimers, classified, as is convention, according to bond length: (peroxide O$_2^{2-}$ ($1.35\,\text{Å} \leq d_{\text{O−O}} < 1.70\,\text{Å}$), superoxide O$_2^-$ ($1.24\,\text{Å} \leq d_{\text{O−O}} < 1.35\,\text{Å}$) and molecular O$_2$ ($d_{\text{O−O}} < 1.24\,\text{Å}$))[51]. Structures containing molecular O$_2$ species are most frequently obtained and are, on average, the most stable, indicating that molecular O$_2$ is the thermodynamic product in charged Li$_2$MnO$_2$F.

To investigate the possible presence of peroxide or superoxide species in charged Li$_2$MnO$_2$F, we performed high-resolution O K-edge RIXS mapping of Li$_{2-x}$MnO$_2$F charged to 5.0 V. In RIXS, incident radiation excites electrons from the O 1s core-level states to empty O 2p valence states, creating core-holes. Relaxation of electrons from the filled O 2p valence states back into the core hole (O 1s), results in emission photons, whose energy are measured, providing a direct probe of the oxygen valence states. The RIXS results are presented in terms of excitation energy, and (emission) energy loss. At an excitation energy of 531 eV, a series of energy loss features near the elastic (zero energy loss) peak can be resolved, arising from transitions to different vibrational energy levels of an O$_2$ molecule (Fig. 3b), with the peak separation of the lowest energy-loss peaks corresponding to the fundamental molecular O$_2$ vibrational frequency ($\nu$) of 1550 cm$^{-1}$ Refs. 14, 16, 52. No vibrational features from peroxide ($\nu = 750$ cm$^{-1}$) or superoxide ($\nu = 1100$ cm$^{-1}$) species are observed. The combined DFT and RIXS results support the idea that molecular O$_2$ is the thermodynamically favoured oxidised O product in highly delithiated Li$_2$MnO$_2$F, and in particular, highlight that molecular O$_2$ is favoured over peroxide and superoxide species.

## Mechanism of O−O dimerisation in delithiated Li$_{2-x}$MnO$_2$F

The structural analysis presented above reveals a strong thermodynamic driving force for O$_2$ formation in highly delithiated Li$_{0.67}$MnO$_2$F, which is consistent with the experimental observation of molecular O$_2$ trapped in the bulk structure[35], and also highlights the necessary role of Mn, O or F migration for this O-redox process to occur. Although this analysis provides valuable thermodynamic insights, it does not give direct information about the atomic mechanisms involved.

We therefore performed ab inito molecular dynamics (AIMD) simulations on a selection of delithiated structures, which allows us to probe their structural evolution as a function of time[53,54]. We examined nine different charged Li$_{0.67}$MnO$_2$F structures, which were obtained as Li$_2$MnO$_2$F from the cluster-expansion, then delithiated (see Methods section). We ran AIMD simulations on each structure at 500 K, a slightly elevated temperature with respect to experiment to allow better sampling of kinetically allowed processes within the accessible simulation timescale (~60 ps). For our detailed analysis here, we focus on one exemplar structure (Fig. 4); structures for the other simulations are shown in the Supplementary Information (Supplementary Fig. 7).

Within this simulation trajectory, three O$_2$ molecules form spontaneously. Further analysis highlights three key points. First, we show that molecular O$_2$ formation is preceded by Mn ion migration (Fig. 4c), with several Mn ions migrating from their initial octahedrally coordinated sites to interstitial sites. These are either also octahedrally coordinated sites, located at the shared edge between two pairs of octahedral sites in the original rocksalt lattice or are fivefold coordinated sites. The displaced Mn ions are stabilised in these interstitial

positions due to large off-site relaxations of the anion sublattice (Supplementary Fig. 8). Second, these Mn migration events leave some O ions in undercoordinated environments (fewer than two Mn nearest-neighbours); a concurrent displacement of the anion sublattice allows some O ions to approach each other which then permits O–O dimer-isation. Third, the mechanism to form molecular $O_2$ involves peroxide and superoxide species appearing as short-lived (picosecond time-scale) reaction intermediates; details of these intermediate $O_2^{n-}$ spe-cies can be resolved by tracking the O–O interatomic separation of the pairs of O that form the $O_2$ molecules (Fig. 4a).

Closer analysis shows that the formation of molecule $O_2(1)$ involves rapid dimerisation (within 2 ps), briefly appearing as a per-oxide intermediate with an O–O bond of 1.40 Å, before detaching from neighbouring Mn ions and relaxing to molecular $O_2$ ($d_{O-O} = 1.22$ Å). $O_2(1)$ can be classified clearly as molecular $O_2$ for the majority of the 60 ps simulation, except for a brief interaction with framework O atoms (Supplementary Fig. 9), which cause temporary (~4 ps) lengthening of the O–O distance to ~1.30 Å (superoxide).

The formation of molecules $O_2(2)$ and $O_2(3)$ shows different behaviour; these dimerise after 3 ps and 7.5 ps respectively, and initi-ally bridge between two or three Mn ions (Supplementary Fig. 5). Molecule $O_2(2)$ initially has a O–O separation of ~1.3 Å, corresponding to a superoxide species, before shortening slightly to an average of 1.25 Å, which is intermediate between the equilibrium superoxide O–O distance and molecular $O_2$ species. Molecules $O_2(2)$ and $O_2(3)$ remain in this intermediate state, until ~55 ps, at which point the O–O dis-tances for both dimers shorten to 1.22 Å (molecular $O_2$), coinciding with each dimer moving away from their neighbouring Mn ions, into a Li-vacancy nanovoid in the structure.

The change in potential energy that accompanies this process of coupled Mn-migration and molecular $O_2$ formation is illustrated in Fig. 4b, which shows DFT energies for selected structures along the AIMD trajectory that were fully relaxed at the hybrid functional level. Each selected structure relaxes to a local energy minimum, and fol-lowing the reaction pathway leads to an overall stabilisation of the system. The superoxide species found in the AIMD simulations are found to be short-lived metastable reaction intermediates on a ps timescale, and exist in a shallow potential well on the energy surface, in agreement with the observation of some superoxide species from the random structure search (Fig. 2). The final product along the AIMD trajectory, containing molecular $O_2$ is confirmed to be the most stable configuration, in agreement with the experimental RIXS data (Fig. 3).

## Charge-discharge process and voltage hysteresis

Understanding the important relationship between voltage hysteresis, TM migration and $O_2$ formation in Li-rich cathodes requires investi-gating both the charge and discharge processes, and considering possible structural changes during the first cycle. Our results show that in $Li_2MnO_2F$, molecular $O_2$ formation at the top of charge drives a reconfiguration of the cathode Mn-host framework. This may then lead to a different reaction pathway and different energetics for discharge compared to charge, and a new structure after the first cycle.

To investigate the role of molecular $O_2$ formation and Mn-migration in voltage hysteresis, we calculated the voltage curve upon charging to $Li_{0.67}MnO_2F$, considering the following two possible alternative end-points (i) metastable structures containing lattice $O^{n-}$ ions, and (ii) a structure containing $O_2$. The structures containing lat-tice $O^{n-}$ ions were obtained by removing Li from the pristine Mn-host framework. The structure containing $O_2$ was obtained by taking the most stable structure containing $O^{n-}$ and performing the minimum number of Mn hops that would leave an O atom undercoordinated and allow $O_2$ to form (Fig. 5). The calculated charge voltage curves were compared with an experimental first charge/discharge curve obtained from GITT measurements (see Methods section). GITT provides a

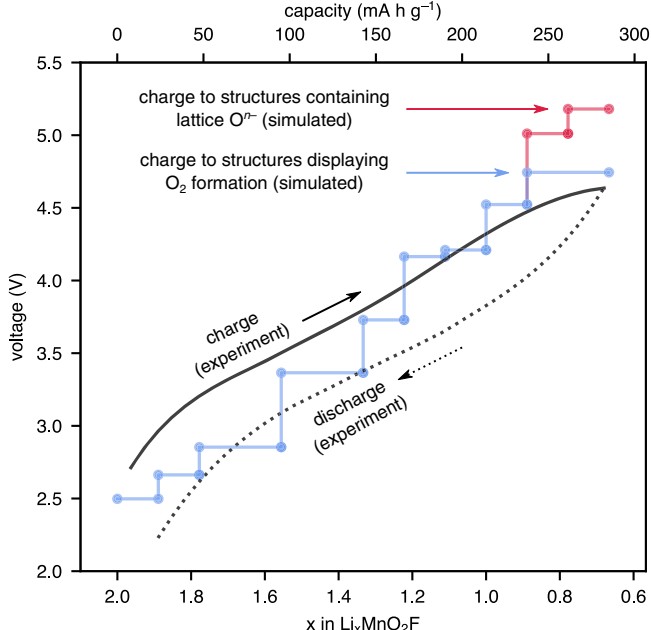

**Fig. 5 | Experimental and calculated voltage curves.** The calculations are based on charging to structures containing lattice $O^{n-}$, or to a structure containing $O_2$, compared against the experimental first-cycle voltage-capacity curve from galva-nostatic intermittent titration technique (GITT) measurements.

voltage profile much closer to the thermodynamic equilibrium than cycling at a conventional C-rate (i.e. 0.1 C).

The results in Fig. 5 reveal two key features. First, the calculated charge voltage curve to structures containing lattice $O^{n-}$ ions exceeds 5 V, which is clearly inconsistent with experiment. Second, the deli-thiated structure containing $O_2$ has a lower predicted voltage, pro-viding a better agreement with experiment. The average calculated charge voltage from the pristine structure to the structure containing $O_2$ is 3.65 V (Supplementary Note S2.3), which compares well with the average experimental voltage of 3.65 V.

We then investigated the discharge process and how this is affected by a structural rearrangement during charge, by re-inserting lithium ions back into the structure containing $O_2$ and calculating the average discharge voltage. Re-inserting Li into the structure containing $O_2$ results in a new discharged structure that contains two $O-Li_6$ environments; the O atoms of the $O_2$ molecule formed during charge are re-incorporated into the lattice as $O^{n-}$ ions. This new discharged structure is significantly higher in energy (~0.4 eV f.u.$^{-1}$) than the pristine $Li_2MnO_2F$ structure, due to the relative instability of the trapped $O-Li_6$ environments (Supplementary Fig. 18). The average calculated discharge voltage is 3.35 V; this is 0.3 V lower than the cal-culated charge voltage, in accord with the -0.16 V hysteresis observed experimentally from a mid-point potential rest experiment (Supple-mentary Fig. 20). In other words, the calculations indicate there is a voltage hysteresis, and this arises from the irreversible structural transformation during charge to form $O_2$.

## Strategies to harness reversible O-redox

Preventing voltage hysteresis and voltage fade is critical for the development of practical O-redox cathodes with high energy densities. Hysteresis can have several sources, including kinetic limitations such as cathode polarisation due to slow lithium diffusion, or first-order phase transitions, both of which should disappear in the limit of extremely slow charging rates, where thermodynamic equilibrium is approached. Another type is 'path-dependent' hysteresis which can arise from irreversible structural changes or slow mobility of host TM cation or anion species[55]. One strategy proposed[18] to suppress

path-dependent hysteresis in O-redox cathodes is to prevent $O_2$ formation by inhibiting TM migration[16,56,57]. In our AIMD simulations, we observe several Mn migration events. Mn migration occurs when Li ions are removed from the structure adjacent to Mn, and the Mn ions then move into new sites, made possible by these vacancies. The relatively large number of Mn ions that migrate is because there are many lithium vacancies at high levels of delithiation in such a lithium-rich system $Li_2MnO_2F$. This implicit link between high Li:TM ratios and ease of Mn migration implies a trade-off in disordered rocksalt cathodes between theoretical capacity and preventing voltage hysteresis. Lower levels of Li-excess, and a more contiguous and connected 3D framework of edge-sharing TM ions is expected to help prevent TM migration.

A similar principle has been proposed in layered Li-rich cathodes, where the superstructure ordering within TM layers affects the stability of those layers; greater stability is achieved with more contiguous TM connectivity[16]. Interesting 3D examples that illustrate this principle include 'partially ordered' spinel-type Mn-oxyfluoride cathodes, which display small voltage hysteresis[38]. These spinel-type materials have a relatively low Li:TM ratio (~1.5 compared with 2 here in $Li_2MnO_2F$), which is likely to have a relatively well-connected network of edge-sharing Mn octahedra. We suggest that these features will tend to reduce the magnitude of off-lattice anion displacements at high states of charge and inhibit TM migration. Achieving a lower level of Li-excess in $Li_2MnO_2F$, if the level of fluorination is kept constant, opens the possibility of using low-valent dopants such as $Mg^{2+}$ and $Zn^{2+[58]}$, in contrast to high-valent $d^0$ dopants such as $Ti^{4+}$ and $Nb^{5+}$ that are often used in disordered rocksalt cathodes[39].

Another approach to the design of Li-rich disordered rocksalt cathodes would be to allow $O_2$ formation and then aim to have fully-reversible TM migration, where the TM ions return to their original sites[18]. By allowing $O_2$ formation, the cathode is rendered much more stable on charge, since metastable lattice $O^{n-}$ species are not trapped in the structure. To allow $O_2$ formation, the roles of local structural rearrangements and reversible transition ion migration become important. The displacement of octahedral cations into tetrahedral sites, as has recently been described for $Cr^{3+}$ to $Cr^{6+}$ in the disordered rocksalt cathodes $Li_{1.2}Mn_{0.2}Ti_{0.4}Cr_{0.2}O_2$ ref. [59] and $Li_2Mn_{0.75}Cr_{0.25}O_2F$ ref. [60], and for Fe ions in $Li_{1.17}Ti_{0.33}Fe_{0.5}O_2$ ref. [15] may permit O–O dimerisation while partially recovering the original structure on discharge. In our study, we show that large off-lattice displacements of anions can permit the migration of Mn ions into interstitial sites. Fully-reversible migration of these Mn ions back to their original sites on discharge could recover the starting structure, and may result in suppressed voltage hysteresis[61].

## Discussion

In conclusion, we have shown that, despite exhibiting limited structural order, Li-rich disordered rocksalt cathodes such as $Li_2MnO_2F$, require transition metal migration before molecular $O_2$ is able to form in the bulk structure on charge. Our results unify the behaviour of disordered rocksalt cathodes with Li-rich ordered layered cathodes, where it is already known that the O-redox process also involves transition metal migration and $O_2$ formation, leading to voltage hysteresis. The nature of hysteresis in disordered materials does differ from the layered systems, since they exhibit solid solution rather than two-phase behaviour during the first charge. Hence, Mn migration and $O_2$ formation occur throughout the rocksalt rather than only in a fraction of the layered cathode that is charged.

Ab initio molecular dynamics combined with DFT calculations show that both TM migration and $O_2$ formation are thermodynamically favoured processes and occur rapidly in charged $Li_2MnO_2F$. High-resolution RIXS mapping confirms that molecular $O_2$ forms (rather than superoxide or peroxide species) and is trapped within lithium vacancy clusters or nanovoids in the bulk material. On discharge, the

$O_2$ molecules are reduced back to $O^{2-}$ lattice ions. When migration is irreversible, these $O^{2-}$ ions occupy sites with a different coordination environment than the pristine compound, which leads to a loss of voltage (-0.3 V) in the first charge/discharge cycle, consistent with the degree of voltage hysteresis observed experimentally. If Mn displacement is fully-reversible, the original $Li_2MnO_2F$ structure and coordination around $O^{2-}$ are recovered, which facilitates O-redox cycling without path-dependent voltage hysteresis. The results presented here suggest that promoting reversible transition metal displacement or suppressing migration altogether in Li-rich disordered rocksalt cathode materials would provide effective strategies to harness O-redox without loss of energy density.

## Data availability

The datasets generated during and/or analysed during the current study are available in the University of Bath repository (https://doi.org/10.15125/BATH-01189).

## Code availability

All code used in this work is described in the Methods section and is openly available.

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

## Acknowledgements

The authors thank the Faraday Institution CATMAT project (EP/S003053/1, FIRG016) and the Henry Royce Institute for financial support. We are also grateful to the HEC Materials Chemistry Consortium (EP/R029431) for Archer high-performance computing (HPC) facilities, GW4 and the UK Met Office for access to the Isambard HPC Service (EP/P020224/1) and for the Faraday Institution's MICHAEL HPC resource. K.M. and A.G.S. thank Dr Stefano Angioni for access to HPC resources through the University of Bath's Cloud Computing Pilot Project. A.G.S. thanks the STFC Batteries Network for an Early Career Researcher Award (ST/R006873/1). This project was supported by the Royal Academy of Engineering under the Research Fellowship scheme. B.J.M. acknowledges support from the Royal Society (URF\R\191006). We acknowledge Diamond Light Source for time on I21 under proposal MM29028-1.

## Author contributions

K.M., B.J.M. and M.S.I. conceived the work. K.M. performed and analysed the calculations with input from A.S. and S.C.; B.J.M. and M.S.I. supervised the computational work. R.A.H. and G.J.R. collected the RIXS measurements and performed the electrochemistry. P.B. supervised the experimental work. K.M., B.J.M. and M.S.I. wrote the manuscript with contributions from all authors.

## Competing interests

The authors declare no competing interests.
