## [Peer Review File · Nature Communications]

Transition metal migration and O₂ formation underpin voltage hysteresis in oxygen-redox disordered rocksalt cathodesReviewer #1 (Remarks to the Author):

The article entitled "Transition metal migration and O₂ formation underpin voltage hysteresis in oxygen-redox disordered rocksalt cathodes" deals with the fundamental understanding of the O-redox activity in disordered rocksalt materials. Since those materials exhibit a smaller first cycle voltage hysteresis than their ordered layered analogous, the understanding of the origin of the more reversible O-redox process is crucial to develop optimized materials. This study is very well conducted by combining different theoretical methods and some experimental work. I believe that the article is suitable for publication in Nature Comm. after some revisions as described below:

- 1) Could the author discuss why they select 2000K for the Monte Carlo simulations to represent to ball milling synthesis conditions. To my point of view the local distribution of Li/Mn around O and F is not representative of the real materials published so far, that, based on PDF and NMR studies, seems more disordered and closer to Monte Carlo simulations using $T = \infty$. The authors should discuss the choice of the 2000K temperature in relation with previous works, since the "constrained Mn" model used to generate deintercalated Li_{0.67}MnO₂F structures could then not be the most realistic one.
- 2) The O-redox mechanism is discussed mainly based on the O-O distance. It will be really nice to confirm the formation of molecular O₂ versus O_n- species also by plotting the DOS or analyze the Bader charges for selected structures.
- 3) Page 7: what is limiting the timescale to 60 ps for the AIMD simulations? A 500K is used here, but far from the room T used for the experimental electrochemical deintercalation. What would be the mechanism then for lower T? Could the authors discuss the activation energy necessary for some Mn migration regarding the exp T° .
- 4) It will be interesting the model again the local distribution of Li/Mn around O and F expected after Mn migration predicted to occur upon cycling.
- 5) In the GITT experiments, the equilibrium is clearly not reached from figure S7. A end of relaxation criteria in dV/dt would have been better than 5h duration.
- 6) It will have been useful to show the 2nd cycling curve while discussing the irreversible character of Mn migration and O₂ formation. According to the discussion it should be lower in voltage.
- 7) Page 12 : the authors claim that if the Mn displacement was reversible, no voltage hysteresis would be observed regarding the O-redox. However this mechanism might be slow and depending on the cycling rate a voltage hysteresis might still be observed. Could the authors include in the discussion this notion of kinetics.

Reviewer #2 (Remarks to the Author):

The authors investigated the oxygen-redox processes in disordered rocksalt cathode (Li₂MnO₂F) using computational and experimental methods. The manuscript reports the atomic-level investigation of the oxygen redox, where the influence of the Mn migration is considered. They showed that the O₂ is thermodynamically favoured compared to superoxide and peroxides, with sufficient evidence to support the claim. The reported results have a sufficient level of significance in the field of disordered cathodes for Li-ion batteries, and the manuscript is written in a clear manner overall. However, some of the claims do not have a sufficient level of evidence to support, as noted in the comments below. The manuscript is suitable for publication in Nature Communications after the authors have addressed the following comments and questions:

Main manuscript:

Page 3:

* A fully random model with $T = \infty$: I suspect that the authors ran an MC simulation with a very high temperature to simulate the infinitely high temperature within the bounds of the lattice model. Although I understand the intention, I suspect that the

distribution in Figure 1b is obtained from an MC run with some very large, yet finite value in practice. Alternatively, the number could have been derived from the analytical calculations. It would be helpful if this point is clarified in the SI (and provide the actual temperature used for the MC run if it was obtained using the first method).

Page 4:

* "Because O–Li6 coordinated oxygen ions appear with a very low frequency, the molecular O₂ that is observed in experiments upon cycling cannot originate uniquely from these O–Li6 oxygen ions.": It is not certain that I completely follow this statement. The fact that the O–Li6 coordination does not happen at full lithiation limit does not mean that these O–Li6 do not form upon some level of delithiation. As far as I can see, the migration of some species (either anionic species or Mn) is likely to happen during charge, leaving room for the rearrangement and the formation of originally-absent O–Li6 coordinations. Could you elaborate on this point?

Page 5:

* The logic behind removing random Li ions from the pristine Li₂MnO₂F structures is unclear. Why didn't the authors remove the least stable Li atoms to be removed from the structures, following the method similar to what was discussed in section 1.6 of SI (i.e., removing the least-stable Li ions based on their electrostatic site energy based on Ewald summation)? While not as accurate as DFT-based assessment, the concept of removing the least-stable Li ions is more justifiable than choosing the random Li ions.

* Since there are 150 configurations used for the "Mn-rearrangement" model while 75 configurations are used for the "constrained-Mn" model, the likelihood of the "Mn-rearrangement" model to have the O–O dimer is higher. While it is true that both 150 and 75 cases are far too small to conclude that the O–O dimerisation takes place in one case but not the other, I suspect that either (1) doubling the number of configurations in the "constrained-Mn" model or (2) removing the least-stable Li ions in the "constrained-Mn" model would indeed reveal some O–O dimerisation.

Page 6:

* It will be helpful to provide a discussion on the underlying reason for the lack of O–O dimerisation in the "constrained-Mn" model.

* "The results therefore indicate a thermodynamic driving force for pristine Li₂MnO₂F to undergo a host framework transformation upon delithiation to allow O–O dimerisation.": I believe that the provided evidence is too weak to make such a definitive statement. The authors compared 150 random configurations ("Mn-rearrangement" model) and 75 configurations by removing random Li ions from the snapshots at T = 2000 K with short-range order ("constrained-Mn" model). However, it seems that the energy range could easily be altered if a different snapshot was taken or different Li ions are removed. This statement warrants a more rigorous study, either by increasing the number of configurations for the "constrained-Mn" model or carefully removing the least-stable Li ions to simulate a more "realistic" case of the delithiation process.

* Regarding the classification of peroxide, superoxide and molecular oxygen, the use of local magnetic moment in addition to the bond length is advised.

Page 7:

* It is not clear how the relative energies in Figure 3a are computed. It would be helpful if the authors explain the method in SI.

* "The structural analysis presented above reveals a strong thermodynamic driving force for O₂ formation in highly delithiated Li_{0.67}MnO₂F, which is consistent with the experimental observation of molecular O₂ trapped in the bulk structure,³⁴ and also highlights the necessary role of Mn migration for this O-redox process to occur.": Regarding the role of Mn migration, the method for classifying the models as the "constrained-Mn" and "Mn-rearrangement" can also be used to argue the impact of O or F migration as the former constrained the positions of Mn, O and F while the latter did not impose any constraint on the positions all three ions. The subsequent MD analysis shows the evolution of Mn, O and F species after the delithiation. Therefore, not sufficient evidence was given on the role of the Mn migration during the delithiation

process.

Page 10:

* "The results in Figure 5 reveal two key features. First, the calculated charge voltage curve to structures containing lattice O²⁻ ions exceeds 5 V, which is clearly inconsistent with the experiment (Figure 5). Second, the delithiated structure containing O₂ has a lower predicted voltage, providing a better agreement with experiment.": It is not too evident how the structures containing O₂ are providing a better agreement with the experiment compared to the structures containing O⁻ ions. The difference between the experimental plot and the red/blue lines seems rather similar/comparable, especially since the inclusion of slightly different structures in constructing the convex hull and the voltage curve could have easily altered the values slightly.

* "The average calculated charge voltage from the pristine structure to the structure containing O₂ is 3.53 V, which compares well with the average experimental voltage of 3.65 V.": It is not clear where these voltages are from (or how they are calculated). Some elaboration would be appreciated.

Page 11:

* "We then investigated the discharge process and how this is affected by a structural rearrangement during charge, by re-inserting lithium ions back into the structure containing O₂ and calculating the average discharge voltage.": The simulation procedure for this is not sufficiently explained. Which site was the Li ion inserted (a random site or most-stable site)? Was it inserted into the relaxed structure or a pristine structure?

* "This new discharged structure is significantly higher in energy (~0.4 eV f.u.⁻¹) than the pristine Li₂MnO₂F structure, due to the relative instability of the trapped O–Li₆ environments (Figure S6).": It is not clear how Figure S6 demonstrates the relative instability of the trapped O–Li₆ environments.

Supplementary Information:

Section 1.2:

* I find it a bit peculiar to set the maximum number of non-zero ECIs to 19. I understand that the main benefit of LASSO is to promote sparsity, which is a desired feature for fast MC simulations. However, the number of resulting ECI should be dictated by the selection that results in the lowest CV score (e.g., assess different values for the hyperparameter and choose the one that results in the lowest CV), rather than restricting the selection scope to an arbitrary number of ECIs. Perhaps I am missing something. Could you provide the justification for restricting the number of ECIs?

* It sounds like the authors are reporting the RMSE value for the entire dataset for the fit. The RMSE does not provide much information regarding the predictive power of the developed CE model. I suggest that the authors provide Leave-One-Out CV and/or k-fold CV values to better assess its predictive ability and to ensure that the model does not suffer from overfitting.

Section 1.3:

* The 6x6x6 cell used for the MC simulation is by no means large enough to capture the long-range effects (e.g., segregation of certain elements due to local ordering). Since the authors already went through the most time-consuming part of developing the accurate CE (assuming that the CV scores are in fact sufficiently low), I suggest that the authors can push an extra mile to run MC on sufficiently large cells (say thousands, tens or even hundreds of thousands of atoms). The associated number of swaps should increase more or less linearly as the number of atoms in the MC cell. However, it should be quite doable as the CE Hamiltonian, as the authors indicated, is very fast.

* "MC-generated structures were analysed using the polyhedral_analysis code": It would be helpful if the authors can specify which for analysis the code was used.

Section 1.4:

* I understand that the SCAN functional is used for better accuracy. However, it is

unclear why the dispersion forces are included in DFT+U and HSE06 but not in SCAN. Could this be elaborated?

General Remark:

*** I suggest the authors provide the DFT/AIMD/CE results in the form of an open repository or as a part of SI for the increased reproducibility of the presented work.**

Reviewer #3 (Remarks to the Author):

This paper describes the study of $\text{Li}_2\text{MnO}_2\text{F}$ by the DFT, AIMD and mXRIS. It reveals that O_2 is thermodynamically favoured over other oxidised O species, and Mn migration happens before O_2 formation.

It is an interesting paper that merits publishing.

There are some questions:

(1) "Discussion" should be changed to "Conclusion".

(2) It is interesting to see the simulations of the second electrochemical cycle to see the behaviour of the Mn^{2+} migration and O_2 formation.

(3) It is suggested to show some ^7Li NMR spectra and Mn synchrotron EXAFS spectra to further confirm the Mn migration and Li migrations.

(4) An interesting question is that Mn migrates at the beginning of the Li-delithiation, and then when $(\text{O}_2)^{\cdot-}$ or O_2 forms? 3.5V? 4.0V?

We are pleased that all the reviewers recommend publication in *Nature Comms* with favourable comments: “This study is very well conducted” (reviewer #1); “sufficient level of significance in the field...the manuscript is written in a clear manner overall” (reviewer #2); “an interesting paper that merits publishing” (reviewer #3).

We have responded point by point to the reviewers’ comments below in blue, and have indicated where changes to the manuscript text have been made.

REVIEWER COMMENTS

Reviewer #1 (Remarks to the Author):

The article entitled “Transition metal migration and O₂ formation underpin voltage hysteresis in oxygen-redox disordered rocksalt cathodes” deals with the fundamental understanding of the O-redox activity in disordered rocksalt materials. Since those materials exhibit a smaller first cycle voltage hysteresis than their ordered layered analogues, the understanding of the origin of the more reversible O-redox process is crucial to develop optimized materials. This study is very well conducted by combining different theoretical methods and some experimental work. I believe that the article is suitable for publication in *Nature Comm.* after some revisions as described below:

1) Could the author discuss why they select 2000K for the Monte Carlo simulations to represent ball milling synthesis conditions. To my point of view the local distribution of Li/Mn around O and F is not representative of the real materials published so far, that, based on PDF and NMR studies, seems more disordered and closer to Monte Carlo simulations using $T = \infty$. The authors should discuss the choice of the 2000K temperature in relation with previous works, since the “constrained Mn” model used to generate deintercalated Li_{0.67}MnO₂F structures could then not be the most realistic one.

We already discuss the choice of Monte Carlo simulation temperature in the Supplementary Information, note 2.1, page 6:

“Li₂MnO₂F is prepared by high-energy ball-milling, and^{27,28} therefore the choice of temperature for the Monte Carlo simulations to model the structure of the as-prepared material is not straightforward. This is due to the unclear and multifaceted relationship between the experimentally used high-energy ball-milling conditions and a thermodynamic “synthetic temperature”. For instance, it has been hypothesised that ball-milling results in local^{29,30} heating or shear-induced reactions. At present, there is no direct method to predict the structures obtained from this complex chemical process. Starting from the work of Kitchaev et al. who propose 1750°C (2023.15 K) as a heuristic boundary for synthetic accessibility by high-energy ball-milling,^{21,31} we decided to use 2000 K to approximate the ball milling synthesis for Li₂MnO₂F, which is produced at a higher ball-milling RPM than similar disordered rocksalt cathodes.”

The reviewer is correct that previously published neutron PDF data [Ref 35, main text] appears to show more random disorder. To show why these previous experiments could not detect the short-range order, we have simulated the X-ray and neutron PDF of Li₂MnO₂F for the 2000K and the $T = \infty$ structures, sampling the same cells that were used to generate the distribution of environments in Figure 1b, Main Text.

Figure S4. Simulated X-ray and neutron PDF patterns for $\text{Li}_2\text{MnO}_2\text{F}$ at the random limit ($T = \infty$ K) and in the 'as-prepared' model ($T = 2000$ K). The simulated data are obtained by averaging over the same set of structures as were used to obtain the distribution of O environments in Figure 1b, Main Text. The 2000K and $T =$ 'infinite' patterns are overlaid and are very similar. The difference patterns are enhanced by 3 and 100 times for the X-ray and neutron patterns respectively. The peaks in the difference pattern are very small. These simulations show why the short-range order, clearly visible in the O-environment analysis in Figure 1b Main Text, cannot easily be observed using either X-ray or neutron total scattering, and is not evident in the previously published PDF data [Ref 35, Main Text].

This new result has been added to the Supplementary Information.

2) The O-redox mechanism is discussed mainly based on the O-O distance. It will be really nice to confirm the formation of molecular O_2 versus O_n^- species also by plotting the DOS or analyze the Bader charges for selected structures.

In line with the reviewer's comments, we have analysed the electronic density of states for the pristine structure (I) and two key structures along the reaction pathway in Figure 4b (structures III and IV), which contain superoxide and molecular O_2 species respectively, and for comparison we have also calculated the density of states for an O_2 molecule isolated in a box. Our results shows that the different O-O species can be clearly differentiated by their density of states. The new results have been added to the Supplementary Information as new Figures S11–S14.

We have performed an analysis of unpaired electron density on specific O atoms from Mulliken charges (which are similar to Bader charges, but are more readily derived for local basis set calculations) for structures I, III and IV. These results show that the different O_2^{n-} species can be differentiated by their unpaired electron density, and the new results have been added to the Supplementary Information as Figure S15.

3) Page 7: what is limiting the timescale to 60 ps for the AIMD simulations? A 500K is used here, but far from the room T used for the experimental electrochemical deintercalation. What would be the mechanism then for lower T ? Could the authors discuss the activation energy necessary for some Mn migration regarding the exp T°.

We first note that our AIMD simulations have not previously been conducted for disordered rocksalt cathodes, and show for the first time the structural transformations in these complex materials. The simulations were conducted for 60 ps, going beyond many similar studies, which are often limited to 10-30 ps [Refs 35, 53, 54, Main Text] as AIMD simulations are highly computationally intensive. Despite this expense, we used a large cell (168 atoms) to reliably capture structural transformations, and furthermore, we ran multiple simulations (8) in parallel, to confirm that similar reaction mechanisms were observed across all cells.

Ideally, simulations would be performed at 300K, but at this temperature, capturing activated events (e.g., transition metal ion migration and O–O dimerisation) would require timescales that are too long (i.e., the calculations are too expensive). Instead, we ran the simulations at a moderately elevated temperature (500K) to capture Mn migration events, which is a standard approach for investigating ion-hopping, even when studying materials with relatively high ion-migration behaviour such as superionic lithium-ion conductors. The assumption made is that the mechanism for diffusion does not change with temperature, which is reasonable in this case. We note that previous AIMD simulations of structural changes in O-redox cathodes (Refs 53, 54, Main Text) have used much higher temperatures (>900K).

To obtain statistically significant results to derive an Mn migration activation energy from an Arrhenius plot, AIMD simulations should obtain a large enough number of ion hops. However, Mn migration does not occur frequently enough to obtain these kinds of results, and it is therefore not realistic to obtain an activation energy for Mn migration from AIMD at present. Nevertheless, the virtue of the AIMD work is that it is able to probe structural dynamics and O₂ formation in a complex, disordered material, where there are too many possible pathways for it to be realistic to systematically compute activation barriers using alternative methods, such as nudged elastic band calculations.

4) It will be interesting the model again the local distribution of Li/Mn around O and F expected after Mn migration predicted to occur upon cycling.

In line with the reviewer's comments, changes to the local distribution of the Li/Mn around O and F upon cycling can be investigated by comparing the distributions in the large AIMD unit cell before and after the AIMD run, during which the structure evolves. The results are shown below and have been added to the Supporting Information, as new Figure S16.

Figure S16. Changes to (a) O- and (b) F-environments in $\text{Li}_2\text{MnO}_2\text{F}$ during cycling. Environments are sampled from the structure undergoing O_2 formation, presented in the AIMD section and Figure 4, Main text. The structure was investigated in the pristine state (before cycling), and fully-discharged state, after the AIMD run, and having been mapped back to a pristine rocksalt lattice (after cycling) (Note S2.4). After cycling, the O-Mn₆ and O-LiMn₅ environments disappear, and the frequency of O-Li₂Mn₄ environments and O-Li₅Mn and O-Li₆ increases, with a small change in the ratio of O-Li₃Mn₃ and O-Li₄Mn₂. In the F-environments, there is a decrease in F-Li₆, and an increase in F-Li₅Mn and F-Li₄Mn₂. The decrease during cycling in F-environments with no transition metal neighbours (F-Li₆) is a feature observed previously for Ni-based oxyfluoride rocksalt cathodes from ^{19}F NMR. [Ref 43 Main Text]

5) In the GITT experiments, the equilibrium is clearly not reached from figure S7. A end of relaxation criteria in dV/dt would have been better than 5h duration.

We agree that equilibrium is not reached after 5 hours. Full equilibration of the electrode can take days making GITT experimentally intractable. However, we performed a longer duration potential rest at the mid-point of the cycle on charge and discharge to examine the equilibrium potential more closely. The difference in the equilibrium potential at this point is 0.16 V. We have added this Figure S20 to the SI.

Figure S20. Mid-point potential rest experiment. Voltage versus capacity plot for $\text{Li}_2\text{MnO}_2\text{F}$ cycled at a rate of 10 mA g^{-1} . Charge in red, discharge in blue. The difference in equilibrium potential at this point is 0.16V .

6) It will have been useful to show the 2nd cycling curve while discussing the irreversible character of Mn migration and O_2 formation. According to the discussion it should be lower in voltage.

The second cycle is now included in the SI as Figure S21 and shows a lower voltage charge, consistent with the irreversible change in structure between the pristine and 1st discharge states.

Figure S21. Load curves for the first and second cycles of $\text{Li}_2\text{MnO}_2\text{F}$.

7) Page 12 : the authors claim that if the Mn displacement was reversible, no voltage hysteresis would be observed regarding the O-redox. However this mechanism might be slow and depending on the cycling rate a voltage hysteresis might still be observed. Could the authors include in the discussion this notion of kinetics.

We have taken the opportunity to clarify our language, in line with the reviewers' constructive comments. The reviewer is correct that 'hysteresis' can arise from polarization of the cathode due to slow Li diffusion kinetics. In the limit of extremely slow charging rates (i.e., approaching

thermodynamic equilibrium), this type of kinetic hysteresis should disappear. We were referring specifically to ‘reaction path hysteresis’, using the definition presented by Van der Ven et al. [Ref 55, Main Text], which is caused by a difference in the chemical reaction pathway for charge and discharge, and can have several origins, including an irreversible change to the cathode structure, or multiple additional mobile species in the cathode (e.g., mobile host-framework cations and anions) that have a much lower diffusivity than Li.

We have modified our language to refer to ‘reaction path hysteresis’ where appropriate and added a note to recognise the difference between polarization and reaction path hysteresis difference to the ‘Discussion’ section.

Reviewer #2 (Remarks to the Author):

The authors investigated the oxygen-redox processes in disordered rocksalt cathode (Li₂MnO₂F) using computational and experimental methods. The manuscript reports the atomic-level investigation of the oxygen redox, where the influence of the Mn migration is considered. They showed that the O₂ is thermodynamically favoured compared to superoxide and peroxides, with sufficient evidence to support the claim. The reported results have a sufficient level of significance in the field of disordered cathodes for Li-ion batteries, and the manuscript is written in a clear manner overall. However, some of the claims do not have a sufficient level of evidence to support, as noted in the comments below. The manuscript is suitable for publication in Nature Communications after the authors have addressed the following comments and questions:

Main manuscript:

Page 3:

1) * A fully random model with $T = \infty$: I suspect that the authors ran an MC simulation with a very high temperature to simulate the infinitely high temperature within the bounds of the lattice model. Although I understand the intention, I suspect that the distribution in Figure 1b is obtained from an MC run with some very large, yet finite value in practice. Alternatively, the number could have been derived from the analytical calculations. It would be helpful if this point is clarified in the SI (and provide the actual temperature used for the MC run if it was obtained using the first method).

The reviewer is correct. We obtained the structures for the “ $T = \infty$ K” by running the MC simulations at 9×10^{14} K. Below, we have confirmed that the results in Figure 1b (plotted as the coloured bars) are almost exactly equal to the derived distribution of environments at a true $T = \infty$ K, (which is a binomial distribution with $n = 6$, $p = 2/3$, plotted in grey) and as such, we have replaced the sampled environments with the derived values in Figure 1b, and modified the Methods section to state that the distribution was obtained in this way.

Page 4:

2) * "Because O–Li₆ coordinated oxygen ions appear with a very low frequency, the molecular O₂ that is observed in experiments upon cycling cannot originate uniquely from these O–Li₆ oxygen ions.": It is not certain that I completely follow this statement. The fact that the O–Li₆ coordination does not happen at full lithiation limit does not mean that these O–Li₆ do not form upon some level of delithiation. As far as I can see, the migration of some species (either anionic species or Mn) is likely to happen during charge, leaving room for the rearrangement and the formation of originally-absent O–Li₆ coordinations. Could you elaborate on this point?

In line with reviewer's comments, we have clarified our language. We were referring to molecular O₂ originating from ions that are in O–Li₆ configurations in the pristine material before charging, and meant to exclude O–Li_x□_{6-x} configurations (where □ is a vacancy) that could form during the charge process. We have modified the text to read:

"Because oxygen ions in O–Li₆ coordination appear with a very low frequency in the pristine material, the molecular O₂ that is observed in experiments upon cycling cannot originate uniquely from starting O–Li₆ oxygen ion sites. Instead, the O₂ molecules must arise from O–Li_nMn_{6-n} (where $n \leq 5$) sites in the pristine material, or O–Li_x□_{6-x} configurations (where □ is a vacancy) that could form during charge due to O, Li or Mn displacement."

Page 5:

3) * The logic behind removing random Li ions from the pristine Li₂MnO₂F structures is unclear. Why didn't the authors remove the least stable Li atoms to be removed from the structures, following the method similar to what was discussed in section 1.6 of SI (i.e., removing the least-stable Li ions based on their electrostatic site energy based on Ewald summation)? While not as accurate as DFT-based assessment, the concept of removing the least-stable Li ions is more justifiable than choosing the random Li ions.

The initial rationale for removing random Li ions was that we observe a redistribution of Li ions from octahedral to tetrahedral sites during relaxation at high levels of delithiation, meaning that the relaxed distribution depends less strongly on the initial distribution. Nevertheless, the reviewer makes a good point that removing Li ions based on electrostatic site energies from an Ewald summation, that we implemented later for the AIMD study, is likely to be an improved strategy.

We have now added 75 more structures to the 'Mn rearrangement' model, using this strategy of removing Li ions based on electrostatic sites energies. The results are shown in the left panel below. Indeed, some of the structures in the electrostatic site removal set (orange) fall lower in energy than structures from the random site set (blue), confirming that it is an improved strategy for sampling low-energy sites. Nevertheless, none of the structures from the new electrostatic site set fall below the energy of the most stable structures containing O₂. Thus, we stress that the conclusions from the results remain the same. We have included the Figure below in the SI as Figure S5, combined the data from the random and electrostatic sites to make an updated version of Figure 2b in the revised manuscript, and updated the Methods.

4) * Since there are 150 configurations used for the "Mn-rearrangement" model while 75 configurations are used for the "constrained-Mn" model, the likelihood of the "Mn-rearrangement" model to have the O-O dimer is higher. While it is true that both 150 and 75 cases are far too small to conclude that the O-O dimerisation takes place in one case but not the other, I suspect that either (1) doubling the number of configurations in the "constrained-Mn" model or (2) removing the least-stable Li ions in the "constrained-Mn" model would indeed reveal some O-O dimerisation.

The rationale for using more structures in the Mn rearrangement model is that the Mn rearrangement model is sampling the entire configurational space for Li/Mn/vacancy and O/F configurations on the rocksalt lattice. That chemical space is much greater than the configurational space for the 'constrained Mn model' which is sampling a more limited number of possible arrangements of the Mn-host framework. We agree, nevertheless, that adding more structures to the 'constrained Mn model' would strengthen the case for O-O dimerisation not occurring spontaneously in that model. Based on the author's suggestion above, we have doubled the number of 'constrained Mn' structures, by adding 75 new structures using the electrostatic site-energy method, as discussed above.

Analysis of the new structures in the 'constrained Mn' model (left panel, above) reveal that none show O-O dimerisation, thus the conclusions from the results remain the same.

Page 6:

5) * It will be helpful to provide a discussion on the underlying reason for the lack of O-O dimerisation in the "constrained-Mn" model.

The lack of dimerisation in the constrained Mn model is because O-O dimerisation is an activated process that requires either Mn-O bond breaking, or Mn/O displacement, and all O atoms in the constrained Mn model start with at least one Mn neighbour. During geometry relaxations in the constrained Mn model, the structures relax to the nearest local energy minimum on the potential energy surface, rather than overcoming the barrier needed for O-O dimerisation. In contrast, in the Mn rearrangement model, some O atoms begin with no Mn neighbours and can dimerise without an activation barrier, so dimers form during the geometry relaxations.

We have added these points to the Main Text, page 6.

6) * "The results therefore indicate a thermodynamic driving force for pristine $\text{Li}_2\text{MnO}_2\text{F}$ to undergo a host framework transformation upon delithiation to allow O–O dimerisation.": I believe that the provided evidence is too weak to make such a definitive statement. The authors compared 150 random configurations ("Mn-rearrangement" model) and 75 configurations by removing random Li ions from the snapshots at $T = 2000$ K with short-range order ("constrained-Mn" model). However, it seems that the energy range could easily be altered if a different snapshot was taken or different Li ions are removed. This statement warrants a more rigorous study, either by increasing the number of configurations for the "constrained-Mn" model or carefully removing the least-stable Li ions to simulate a more "realistic" case of the delithiation process.

We recognise that a definitive statement should not be made based on the data provided.

In line with the reviewer's comments, we have taken two steps: firstly, adding more structures to the search, to strengthen the evidence, described above for comments 3) and 4) (75 structures to the 'constrained' Mn model. Secondly, we have tempered the language regarding the thermodynamics, from 'indicate' to 'suggest ... may exist':

Page 6

"The results therefore suggest a thermodynamic driving force may exist for pristine $\text{Li}_2\text{MnO}_2\text{F}$ to undergo a host framework transformation upon delithiation to allow O–O dimerisation"

7) * Regarding the classification of peroxide, superoxide and molecular oxygen, the use of local magnetic moment in addition to the bond length is advised.

As requested, we have performed this analysis (presented below) for the structures from the Mn-rearrangement model. We can confirm that the bond length classification corresponds well to the expected magnetic moment for that type of O–O dimer. The results have been added to the Supporting Information as new Figure S6.

Page 7:

8) * It is not clear how the relative energies in Figure 3a are computed. It would be helpful if the authors explain the method in SI.

The energies were calculated relative to the most stable configuration from the entire search. Thus, zero energy corresponds to the most stable obtained from the Mn rearrangement model. This detail has been clarified in the caption of Figure 3a, and in Supplementary Note S2.2.

9) * "The structural analysis presented above reveals a strong thermodynamic driving force for O_2 formation in highly delithiated $Li_{0.67}MnO_2F$, which is consistent with the experimental observation of molecular O_2 trapped in the bulk structure,³⁴ and also highlights the necessary role of Mn migration for this O-redox process to occur.": Regarding the role of Mn migration, the method for classifying the models as the "constrained-Mn" and "Mn-rearrangement" can also be used to argue the impact of O or F migration as the former constrained the positions of Mn, O and F while the latter did not impose any constraint on the positions all three ions. The subsequent MD analysis shows the evolution of Mn, O and F species after the delithiation. Therefore, not sufficient evidence was given on the role of the Mn migration during the delithiation process.

This is a good point, and we agree that the 'Mn rearrangement' model is not sufficient evidence for Mn migration being necessary for the O-redox process; the behaviour could be alternatively attributed to O or F migration. We have modified the text on Page 8 to recognise this:

"The structural analysis presented above reveals a strong thermodynamic driving force for O_2 formation in highly delithiated $Li_{0.67}MnO_2F$, which is consistent with the experimental observation of molecular O_2 trapped in the bulk structure,³⁴ and also highlights the necessary role of Mn, O or F migration for this O-redox process to occur."

Page 10:

10) * "The results in Figure 5 reveal two key features. First, the calculated charge voltage curve to structures containing lattice O_n^- ions exceeds 5 V, which is clearly inconsistent with the experiment (Figure 5). Second, the delithiated structure containing O_2 has a lower predicted voltage, providing a better agreement with experiment.": It is not too evident how the structures containing O_2 are providing a better agreement with the experiment compared to the structures containing O_n^- ions. The difference between the experimental plot and the red/blue lines seems rather similar/comparable, especially since the inclusion of slightly different structures in constructing the convex hull and the voltage curve could have easily altered the values slightly.

Motivated by the reviewers' constructive comment, we have considered in greater detail how the structures we have used might relate to the real materials after different stages of cycling. To calculate the voltage curve for O_2 formation, we initially used the energy of the most stable structure from the 'Mn rearrangement' search, which puts no constraints on Mn/O/F movement (i.e., from a global ground-state search). We now reason that this structure is most representative of a thermodynamic reaction product.

Structural changes in the cathode during early cycles, however, will initially be governed by kinetics. To obtain a representative structure for a kinetic product, we took the most stable charged structure containing O_n^- ions, and established the minimum number of Mn migration steps that would allow an O_2 molecule to form. We relaxed this new structure and found it to be more stable than the structures containing O_n^- ions, yet with a higher energy than the structure containing O_2 from the Mn rearrangement search. The computed voltage to this new structure provides a close match with the experimental first-cycle charge curve.

We suggest that the new structure containing O_2 obtained by Mn swaps is more representative of a kinetic product, formed after the end of the first charge, by only a few transition metal ion migration events. In contrast, the more-stable structure containing O_2 from the random (global) search is likely to be more representative of a thermodynamic product, formed after multiple cycles, each of which involves progressive transition metal ion migration. We have replaced the calculated voltage curve (Figure 5, Main text) with a version where the structure containing O_2 is representative of the first-cycle product, updated the calculated average charge and discharge voltages, added the new data point to the convex hull plot (Figure S17), and have elaborated on this new result in the caption of Figure S17.

11) * "The average calculated charge voltage from the pristine structure to the structure containing O₂ is 3.53 V, which compares well with the average experimental voltage of 3.65 V.": It is not clear where these voltages are from (or how they are calculated). Some elaboration would be appreciated.

These calculated intercalation voltages were determined directly from the Nernst equation, according to the equation in Section S2.3. Because we do not calculate the stepwise discharge process, the average voltages were calculated as the open-circuit voltage between the relevant structures at $x = 2.0$ and $x = 0.667$ in $\text{Li}_x\text{MnO}_2\text{F}$. Such an approximation is valid in this case, because the disordered cathode displays a smooth, sloping voltage profile. We have clarified this detail to the Supplementary Information, note S2.3. The average experimental voltage was determined from the GITT measurements.

Page 11:

12) * "We then investigated the discharge process and how this is affected by a structural rearrangement during charge, by re-inserting lithium ions back into the structure containing O₂ and calculating the average discharge voltage.": The simulation procedure for this is not sufficiently explained. Which site was the Li ion inserted (a random site or most-stable site)? Was it inserted into the relaxed structure or a pristine structure?

To generate the discharged structures, we used structure mapping using the *map_structure_to_reference* function in the ICET code, performed on relaxed, highly delithiated structures. This function returns a structure with ions mapped onto the nearest lattice positions permitted in the cluster expansion basis (including dummy atoms for vacancy sites). Since our basis was for the discharged structure where cations are only permitted on octahedral sites; Li-ions in tetrahedral sites were mapped back to the nearest octahedral site. We then inserted Li ions into all the remaining vacancy sites identified in the structure mapping and relaxed the resulting structures.

We have added this into the Supplementary Information as new Supplementary Note S2.4.

13) * "This new discharged structure is significantly higher in energy (~ 0.4 eV f.u. $^{-1}$) than the pristine Li₂MnO₂F structure, due to the relative instability of the trapped O–Li₆ environments (Figure S6).": It is not clear how Figure S6 demonstrates the relative instability of the trapped O–Li₆ environments.

We appreciate this comment, and agree that the data presented in the calculated convex hull shows only that the structure containing O–Li₆ configuration is higher in energy, and it does not provide evidence that the high energy can be attributed to the O–Li₆ sites.

We had initially inferred that O–Li₆ sites were high-energy from their near-complete absence in the Monte Carlo simulations below $T = 5000$ K (Supplementary Figure S1). To establish a more direct association between the energy of the O–Li₆ sites and the high energy of the discharged structure, we have calculated the sites energies of O ions using electrostatics for each of the five pristine structures, and for the discharged structure containing O–Li₆, presented below.

The lower panels show the site energies from electrostatics for different O coordination environments, showing a trend of higher energy with increasing coordination to Li ions. In the pristine structures, all O atoms are coordinated to one or more Mn ions, and the relative energy of each structure is no greater than 0.1 eV f.u. $^{-1}$ of the most stable structure. In the discharged structure, there are two O–Li₆ environments, which do not appear in any of the other structures and are the highest energy O sites across all structures. The discharged structure has an energy of +0.4 eV f.u. $^{-1}$. We can therefore confirm that i) the O–Li₆ sites are high-energy and ii) the high energy of the O–Li₆ sites contributes to the high energy of the discharged structure, relative to the pristine structures.

We have added this analysis to the Supporting Information as Figure S18.

Supplementary Information:

Section 1.2:

14) * I find it a bit peculiar to set the maximum number of non-zero ECIs to 19. I understand that the main benefit of LASSO is to promote sparsity, which is a desired feature for fast MC simulations. However, the number of resulting ECI should be dictated by the selection that results in the lowest CV score (e.g., assess different values for the hyperparameter and choose the one that results in the lowest CV), rather than restricting the selection scope to an arbitrary number of ECIs. Perhaps I am missing something. Could you provide the justification for restricting the number of ECIs?

We recognise that the description we used was not full or clear, and we would like to elaborate on the fitting method. In the fitting, we used a recursive feature elimination (RFE) approach, in combination with the LASSO cross-validation estimator. In the RFE procedure, first, the estimator is trained on the initial set of effective cluster interactions (ECIs) and the importance of each ECI is determined, along with a cross-validation score. Then, the least important ECI is pruned from the current set, and the procedure is recursively repeated on the pruned set until either i) a selected number of ECIs are obtained or ii) the cross-validation score no longer improves, giving a resulting number of ECIs. We used approach ii), i.e., no upper limit was set for the number of ECIs, and instead, the optimal number of ECIs (in this case, 19) was determined automatically by the cross-validation score.

We have clarified these details in the Methods section of the Supplementary Information:

“The ECIs were obtained using a least absolute shrinkage and selection operator (LASSO) regression analysis, with a recursive feature elimination (RFE) approach, in which minimally contributing parameters are removed recursively, and a cross-validation score calculated, repeated until the cross-validation score no longer improves. The LASSO + RFE approach resulted in a fit with 19 non-zero ECIs”

15) * It sounds like the authors are reporting the RMSE value for the entire dataset for the fit. The RMSE does not provide much information regarding the predictive power of the developed CE model. I suggest that the authors provide Leave-One-Out CV and/or k-fold CV values to better assess its predictive ability and to ensure that the model does not suffer from overfitting.

We would like to clarify the methods used. Here, the RMSE refers to a k-fold cross-validation (CV) RMSE, obtained by the RFE approach, by repeatedly sampling training and validation sets to ensure the model does not suffer from overfitting. As suggested below, we are providing all the computational data, including the CE fitting, in an open-source repository. We have improved the description of the fitting in the Methods section of the Supplementary Information.

Section 1.3:

16) * The 6x6x6 cell used for the MC simulation is by no means large enough to capture the long-range effects (e.g., segregation of certain elements due to local ordering). Since the authors already went through the most time-consuming part of developing the accurate CE (assuming that the CV scores are in fact sufficiently low), I suggest that the authors can push an extra mile to run MC on sufficiently large cells (say thousands, tens or even hundreds of thousands of atoms). The associated number of swaps should increase more or less linearly

as the number of atoms in the MC cell. However, it should be quite doable as the CE Hamiltonian, as the authors indicated, is very fast.

As requested by the reviewer, we have performed these additional MC simulations for $\text{Li}_2\text{MnO}_2\text{F}$ at 2000K using supercells of $9\times 9\times 9$ (1,458 atoms) and $18\times 18\times 18$ (11,664 atoms). We analysed the cells by characterising i) the distribution of O environments, in the same manner as the analysis in Figure 1, Main Text. We also assessed the possibility of lithium clustering (i.e., phase-segregation-type behaviour) by calculating the number of Li-centered octahedra that edge-share with other Li octahedra. A large increase in Li-Li edge sharing octahedra relative to the $6\times 6\times 6$ cell would indicate significant Li clustering and phase-segregation in the larger cells.

The O environments and Li-Li edge sharing octahedra analysis for the $9\times 9\times 9$ and $18\times 18\times 18$ cells have been added as new Figures S2 and S3. Relative to the the distributions from the $6\times 6\times 6$ cell, the O-environment frequencies show a maximum difference of $\sim 5\%$ and the Li-Li edge sharing octahedra have a maximum difference of $< 2\%$, indicating that the results do not change significantly in the larger cells. We ran Monte Carlo simulations for $33\times 33\times 33$ unit cells (71,874 atoms), but our code to analyse polyhedral connectivity was unable to handle cells of this size, consistently causing memory overloads, rendering these large cells computationally intractable at this time.

Nevertheless, the consistency across the $6\times 6\times 6$, $9\times 9\times 9$ and $18\times 18\times 18$ cells mean that we are confident that the larger cells do not display significant segregation of elements that would affect the results. Importantly, O-Li₆ environments also do not appear in appreciable quantities in the larger cells, meaning that the conclusion that Mn, O or F migration is required for O₂ formation, remains the same. We have updated the Methods to note the MC calculations in the larger cells.

17) * "MC-generated structures were analysed using the polyhedral_analysis code": It would be helpful if the authors can specify which for analysis the code was used.

The *polyhedral_analysis* code was used to analyse the local environments around specific O and F polyhedra, to produce Figure 1b. The code was also used in several other cases to identify polyhedral environments, such as for the analysis of the site energies from electrostatics for O atoms in different environments (in response 13) and the Li-Li edge-sharing neighbours. These details have been added to Supplementary Information Note 1.7.

Section 1.4:

18) * I understand that the SCAN functional is used for better accuracy. However, it is unclear why the dispersion forces are included in DFT+U and HSE06 but not in SCAN. Could this be elaborated?

Dispersion forces are completely missing from DFT+U and HSE06 functionals, so were included using the D3 scheme because we anticipated the significance of dispersion in describing the interaction between O₂ molecules and their surrounding environment in the charged cathode.

Dispersion terms were not explicitly added to the SCAN calculations because the standard parameterisation of the SCAN functional achieves an effective description of intermediate-

range van der Waals forces (the intermediate range is defined as roughly the distance between nearest neighbour atoms at equilibrium) as discussed by [J. Sun et al. Nature Chemistry, 8, 831–836 (2016)]. Such an approach has been found to be suitable for describing the behaviour of O₂ molecules trapped within cathodes and has been used as a standard for investigation O-redox cathodes previously [Ref 51, Main Text]. We have added to the Methods to note this point.

General Remark:

19) * I suggest the authors provide the DFT/AIMD/CE results in the form of an open repository or as a part of SI for the increased reproducibility of the presented work.

As suggested, we are preparing the entire computational dataset, including notebooks to analyse the data and generate the Figures, to be deposited in the University of Bath open-access public repository when the manuscript is accepted.

Reviewer #3 (Remarks to the Author):

This paper describes the study of $\text{Li}_2\text{MnO}_2\text{F}$ by the DFT, AIMD and mXRIS. It reveals that O_2 is thermodynamically favoured over other oxidised O species, and Mn migration happens before O_2 formation. It is an interesting paper that merits publishing. There are some questions:

(1) "Discussion" should be changed to "Conclusion".

This section is labelled 'Discussion' because the Nature Communications format does not include a 'Conclusion' heading – we refer this comment to the editors.

(2) It is interesting to see the simulations of the second electrochemical cycle to see the behaviour of the Mn^{2+} migration and O_2 formation.

We agree with the reviewer that this is an interesting question and warrants future investigation. Nevertheless, modelling structural transformations during a specific number of cycles (i.e., two), is far beyond the scope of even the state-of-the-art techniques used in this study. At present, it is only possible to simulate structures formed during the first/early cycles (kinetic products – from AIMD) and the products of 'long-term' cycling (closer to a thermodynamic ground state – from structure-searching).

(3) It is suggested to show some ^7Li NMR spectra and Mn synchrotron EXAFS spectra to further confirm the Mn migration and Li migrations.

Operando Mn K-edge EXAFS data has been previously published by our groups in JACS 2020 [Ref 35, Main Text], which shows that the octahedral coordination of the Mn ions is broadly maintained throughout the first cycle, consistent with our new results here, showing migration of Mn to new octahedral (rather than tetrahedral) sites. The Mn K-edge EXAFS displays some weak changes in the region corresponding to the Mn-Mn first-neighbour distance, which may suggest changes to the Mn host framework (i.e., Mn migration), but the disordered nature of the materials makes this interpretation very challenging.

Regarding lithium migration, mobility of Li using solid state NMR requires either pulsed-field gradient (PFG) NMR or relaxometry over multiple temperatures to achieve Arrhenius relationships. PFG is not possible in paramagnetic systems as the T_2 (NMR linewidth) is too short meaning the signal does not survive the PFG experiment. Likewise, relaxometry over temperature would be dominated by the paramagnetic effects, and cannot resolve Li mobility. The mobility of Li, however, is not a central topic of the present study, where we focus on Mn migration and O–O dimerization in relation to voltage hysteresis.

(4) An interesting question is that Mn migrates at the beginning of the Li-delithiation, and then when (O_2)ⁿ⁻ or O_2 forms? 3.5V? 4.0V?

We agree that this is a very interesting question. Nevertheless, determining specifically when Mn migration is initiated, and when O–O dimerisation can first occur during charge will require a similar level of study as has been performed in the current work for the top of charge, (i.e. a full structural search and AIMD simulations) but at multiple compositions along the charge

curve. A many-fold increase in the number of calculations is beyond the scope of the present work and will be the topic of future investigations.

Reviewer #1 (Remarks to the Author):

In the revised version of the article entitled "Transition metal migration and O₂ formation underpin voltage hysteresis in oxygen-redox disordered rocksalt cathodes", the authors took into consideration all remarks of the three reviewers. Additional answers are also provided in the rebuttal letter to clarify some points. The quality of the paper has notably improved. I believe that the fundamental understanding of the O-redox activity in Li₂MnO₂F studied in this paper with a proper methodology, is crucial for developing optimized materials and will be of wide interest to the community.

I recommend the publication of the paper in its present form in Nature Communications.

Reviewer #2 (Remarks to the Author):

The authors have addressed all of my comments and questions in sufficient detail. I spotted one minor mistake for comment #6, where the authors added "suggest" and "may exist" in the response letter, but forgot to add "may exist" in the manuscript itself. Other than this minor mistake, I believe that the manuscript is suitable for publication in Nature Comm.

Reviewer #3 (Remarks to the Author):

It is satisfactory now.

Response

We thank the reviewers for their constructive comments, and the opportunity to strengthen the manuscript.

Reviewer #2 (Remarks to the Author):

The authors have addressed all of my comments and questions in sufficient detail. I spotted one minor mistake for comment #6, where the authors added "suggest" and "may exist" in the response letter, but forgot to add "may exist" in the manuscript itself. Other than this minor mistake, I believe that the manuscript is suitable for publication in Nature Comm.

Thanks – this has now been added.